# *Bothrops moojeni* Venom and Its Components Strongly Affect Osteoclasts’ Maturation and Protein Patterns

**DOI:** 10.3390/toxins13070459

**Published:** 2021-06-30

**Authors:** Fernanda D’Amélio, Hugo Vigerelli, Álvaro Rossan de Brandão Prieto-da-Silva, Eduardo Osório Frare, Isabel de Fátima Correia Batista, Daniel Carvalho Pimenta, Irina Kerkis

**Affiliations:** 1Laboratory of Genetics, Butantan Institute, São Paulo 05503-900, Brazil; fernanda.damelio@esib.butantan.gov.br (F.D.); alvaro.prieto@butantan.gov.br (Á.R.d.B.P.-d.-S.); eduardo.frare@butantan.gov.br (E.O.F.); 2The Postgraduate Program in Toxinology, Butantan Institute, São Paulo 05503-900, Brazil; 3Centre of Excellence in New Target Discovery (CENTD), Butantan Institute, São Paulo 05503-900, Brazil; isabel.batista@butantan.gov.br; 4Laboratory of Biochemistry and Biophysics, Butantan Institute, São Paulo 05503-900, Brazil; dcpimenta@butantan.gov.br

**Keywords:** osteoclasts, differentiation cellular, toxins, *Bothrops moojeni*

## Abstract

Osteoclasts (OCs) are important for bone maintenance, calcium balance, and tissue regeneration regulation and are involved in different inflammatory diseases. Our study aimed to evaluate the effect of *Bothrops moojeni’s* venom and its low and high molecular mass (HMM and LMM) fractions on human peripheral blood mononuclear cell (PBMC)-derived OCs’ in vitro differentiation. *Bothrops moojeni*, a Brazilian lanced-head viper, presents a rich but not well-explored, venom composition. This venom is a potent inducer of inflammation, which can be used as a tool to investigate the inflammatory process. Human PBMCs were isolated and induced to OC differentiation following routine protocol. On the fourth day of differentiation, the venom was added at different concentrations (5, 0.5, and 0.05 µg/mL). We observed a significant reduction of TRAP+ (tartrate-resistant acid phosphatase) OCs at the concentration of 5 µg/mL. We evaluated the F-actin-rich OCs structure’s integrity; disruption of its integrity reflects bone adsorption capacity. F-actin rings phalloidin staining demonstrated that venom provoked their disruption in treated OCs. HMM, fraction reduces TRAP+ OCs at a concentration of 5 µg/mL and LMM fraction at 1 µg/mL, respectively. Our results indicate morphological changes that the venom induced cause in OCs. We analyzed the pattern of soluble proteins found in the conditioned cell culture medium OCs treated with venom and its fractions using mass spectrometry (LC-MS/IT-Tof). The proteomic analyses indicate the possible pathways and molecular mechanisms involved in OC reduction after the treatment.

## 1. Introduction

Belonging to the Viperidae family, *Bothrops moojeni* is responsible for the most snakebite accidents, becoming a public health problem in Brazil and a group of larger importance that causes bothropic accidents [1]. Being the most complex venom of the animal kingdom, snake venom contains actives molecules with potential pharmacological effects [2]. *B. moojeni* venom, in particular, is characterized as possessing activated proteolytic, coagulant, and hemorrhagic factors, composed of metalloproteases (SVMPs—snake venom metalloproteases), serine proteases, phospholipases, and L-amine oxidase acid [3]. Comprehension of the whole venom and individual molecules function is important for the evaluation of their mechanism action on biological systems. 

Osteoclasts (OCs) are giant multinucleated cells, acting in bone tissue reabsorption and calcium metabolism. Being essential for homeostasis, OCs’ differentiation occurs from mononuclear precursors in the bone marrow [4]. The regulation of OCs formation and differentiation is orchestrated by cytokines and factors, such as RANKL (ligand cytokine for nuclear activation of the ĸB factor); colony stimulating factor one (CSF-1); TNF-α (tumor necrosis factor-alpha); interleukin (IL) 1, IL 6, IL 11, and IL 17; M-CSF (macrophage colony-stimulating factor); and prostaglandin, among others [5]. RANKL and CSF-1 are also necessary for inducing the expression of genes that genetically typify the OCs lineage, such as tartrate-resistant acid phosphatase (TRAP) [6,7,8,9]. 

The bone tissue remodeling process occurs with a continuous dynamic balance between bone formation, performed by osteoblasts, and bone absorption, performed by OCs [10]. Functional disorders involving these cells, especially those related to excessive OCs bone resorption activity, are present in bone and joint diseases, such as rheumatoid arthritis, osteoporosis, Paget’s disease, and osteosarcoma [11,12]. 

On the one hand, the use of an OCs in vitro model is necessary to elucidate the mechanisms and pathways that can be affected by the crude venom or its components during these cells’ differentiation. Moreover, such studies allow a better understanding of bioactive molecules’ mechanisms of action, which compose the venoms. They help unveil these molecules’ action on OCs formation and function and point out new possible therapeutic targets. To date, no studies have evaluated the influence of *B. moojeni* venom and its components on human OCs’ differentiation. 

The present study’s main goal was to evaluate the effect of *B. moojeni* venom and its low and high molecular mass (LMM and HMM) fractions on OCs differentiation and maturation. We also performed secretome and pathway analysis of mature OCs, which enabled us to carve out the secreted protein composition changes induced by *B. moojeni* venom and its components in mature OCs. Previous results of this work have been published in the 1st International Electronic Conference on Toxins 2021 [13].

## 2. Results and Discussion 

### 2.1. Effect of B. moojeni Crude Venom on Cell Viability, TRAP+ OCs Number, and F-Acting Ring Integrity 

Previous studies have showed the effects of snake venoms in OC differentiation. For instance, a hemodynamic disintegrin called contortrostatin, derived from the venom of the snake *Agkistrodon contortrix,* presented itself as a potent inhibitor of the differentiation of neonatal osteoclasts in rats [14]. Besides, ecystatin, analogous to the peptide isolated from the snake venom *Echis carinatus*, has a different effect on integrin αVβ3, causing a decrease in OCs’ multinucleation formation, probably being involved in cell migration and adhesion [15]. Therefore, studies on new therapeutic targets that inhibit osteoclasts’ formation, impairing their function, are extremely important for new treatments of great socio-economic value [10]. 

The effect of *B. moojeni* venom in an OCs differentiation model was evaluated using phenotypic assays based on the characteristics of mature OCs, such as the number of TRAP+ cells, F-acting ring integrity, and OCs multinuclearity. To evaluate the toxic effect of *B. moojeni* venom on OCs, we performed a mature OCs viability test on day 15 of differentiation. For this purpose, differentiation into OCs was induced using RANKL immediately after PBMC plating. The venom was added at different concentrations (5, 0.5, and 0.05 µg/mL) on day 4 after plating, and it was maintained until before the end of differentiation (day 15). The CCK8 method was adopted to evaluate OCs’ primary culture viability based on hydrogenase activity measurement. For this, the absorbance value was reversed in the percentage of living cells. According to Figure 1A, no statistically significant difference in cell viability was observed in the OCs culture at all tested concentrations. 

TRAP is a specific marker of mature OCs; therefore, we performed TRAP staining at the end of the PBMC differentiation protocol in the groups treated with crude venom at the same concentrations used in the viability assay. Besides, this staining was performed in two control groups, one with PBMC induced for differentiation and the other with PBMC in the basal medium. TRAP staining demonstrated, in the positive control, multinucleated and active OCs appear in a purple color, where it is possible to observe the stained nuclei. Cells not able to metabolize become very dark in color (Figure 1B–E). Figure 1B demonstrates the TRAP+ OCs control culture and TRAP+ OCs treated with crude venom at a concentration of 0.05 (Figure 1C), 0.5 (Figure 1D), and 5 µg/mL (Figure 1E). The venom treatment provides a hard effect on OC morphology. OCs in positive control demonstrate a “spread out” morphology with clearer cytoplasm and definite multiple nuclei. In contrast, according to the venom concentration increase, OCs exhibit shrunken cytoplasm and very dark TRAP+ staining, as shown in Figure 1E. Figure 1(B1,E1) demonstrate OCs stained with phalloidin, which helps to realize the shrunken cytoplasm in venom-treated OCs. A similar effect was observed in OCs of rats treated with estrogens [16], including cimetidine and eupatilin [17].

Next, we counted the number of TRAP+ OCs in OCs of the positive control in OCs treated with venom. The results showed that the treatment with 5 µg/mL of venom significantly reduces the number compared to the positive control (Figure 1F). A mixed population of bone marrow precursors was used in an OCs in vitro cell model; therefore, on day 15, we observed OCs surrounded by differentiated immune and several fibroblastic cells. OCs differentiation can be divided into three stages: first, OCs precursor generation from day 0 until the third day; second, immature fused polykaryons formation from the third day until approximately the sixth day; and third-day multinucleated OCs formation/maturation from day 6 until the 15th day. The viability test suggests that we did not observe cell death. However, Figure 1A,F indicates that OCs differentiation was inhibited right after venom addition, especially at the concentration of 5 µg/mL, compared with other concentrations. Thus, this prevented further uncommitted precursor differentiation to OCs occurring among the bone marrow precursors in the first stage [18,19]. 

OCs are bone-resorbing cells acting as fundamental mediators of bone conditions. Mature OCs polarize and reorganize their cytoskeleton to create an F-actin-rich ring upon adhesion to the bone [20]. Staining of F-actin rings with phalloidin allowed us to observe the conservation and integrity of these structures. The OCs treated with venom show a difference in the integrity of the ring (Figure 1G,H). Figure 1G demonstrates intact F-actin ring formation in the positive control. After OCs were treated with different venom concentrations, the rings’ gradual disruption was observed, which depends on the venom concentration. Figure 1H shows that OCs demonstrate the intact ring on one side, while the ring shows topic disruptions on the other side. This effect is stronger at a concentration of 0.5 μg/mL (Figure 1I), and in Figure 1J, a destroyed F-actin ring is shown. The F-actin-rich ring morphology complements our data on the viability and Trap+ tests provide insight into the venom’s effect, suggesting that venom-treated OCs are not able to metabolize since we observed strong ring disruption in OCs treated with 0.5 and 5 μg/mL of venom [20].

### 2.2. Effect of Low and High Molecular Mass Fraction of B. moojeni on Viability and TRAP+ OCs Number, and F-Acting Ring Integrity 

To refine the study venom effect in the OC model, we treated mature OCs with fractions of LMM and HMM fractionated by cutting membrane at 10 kDa at concentrations of 5, 1, and 0.5 µg/mL, where high molecular mass components are greater than 10 KDa and low mass components are less than 10 KDa. The treatment with LMM and HMM fractions showed no toxic effect on OC viability (Figure 2A). 

TRAP staining revealed TRAP+ OCs in the positive control, and OCs treated with LMM and HMM fraction. However, a morphological difference was observed in OCs treated with fractions that show small-multinucleated OCs with less cytoplasm that is morphologically different with OCs from the positive control.

The number of TRAP+ OCs decreased significantly in the groups treated with high molecular mass (HMM) (5 µg/mL) and low molecular mass (LMM) (1 µg/mL) fractions when compared to a positive control (Figure 2F). Notably, the HMM and LMM fractions demonstrate the dose-dependent effect of the OCs TRAP+ cells number. This HMM provides a stronger effect at 5 µg/mL. This effect decreases at 1 and 0.5 µg/mL, respectively. Interestingly, the LMM fraction showed the opposite effect to the HMM fraction. LWM provides a stronger effect at 1 and 0.5 µg/mL, respectively, while at the 5 µg/mL venom concentration, the number of TRAP+ cells was comparable with the positive control. Regarding OCs differentiation, HMM and LMV fractions did not induce cell death at day 15 (Figure 2A); however, inhibition of OCs precursor differentiation (stage 1) was observed in different concentrations. For HMM, the strongest inhibition occurred at 5 µg/mL; for LMM, it occurred at 1 and 0.5 µg/mL as TRAP staining revealed [18]. 

Figure 2C–F shows TRAP+ OCs in the positive control and OCs treated with LMM and HMM fractions. It demonstrates TRAP+ cells in the positive control (Figure 2C) and after the treatment with the higher dose of *B. moojeni* venom (Figure 2D), with the highest dose of HMM fraction (Figure 2E) and with the LMM dose (Figure 2F), which provide statistical significance for the TRAP+ cell in comparison with the positive control (Figure 2B). However, a morphological difference was observed in OCs treated with venom and their fractions, which showed OCs with shrunken cytoplasm. This effect seemed stronger in OCs treated with HMM fraction, while venoms and LMM fraction showed a similar pattern. 

In the positive control, the F-actin ring’s formation was intact (Figure 2G, arrow). The groups treated with crude venom, HMM, and LMM fractions differ from the positive control. The group incubated with crude venom demonstrated a lack of F-actin ring structure formation (Figure 2H, arrowhead). The groups treated with HMM and LMM fractions showed disruption of the F-actin rings or a “ghost-like” morphology (Figure 2I,J, arrowhead). No formation of F-actin-rich rings was observed after the treatment with LMM fraction. The F-actin rings were strongly affected by the HMM and LMM fraction treatments, indicating possible metabolic collapse of OCs. 

### 2.3. Cell Culture Medium Soluble Protein Analysis of Mature OCs (Day 15) 

For this study, we considered secretome as all the proteins found in the supernatant analysis of mature OCs. This analysis was performed using liquid chromatography coupled to mass spectrometry to identify proteins specific for mature OCs treated with *B. moojeni* crude venom (5 µg/mL), HMM (5 µg/mL), and LMM (1 µg/mL) fractions versus proteins found in the positive or negative control (Table 1). The table containing all identified protein presented in Appendix A.

We identified 120 proteins for the positive control, 53 proteins for the negative control, 103 proteins for the group treated with *B. moojeni* crude venom, 22 proteins for the HMM group, and 22 proteins for the LMM group. It is noteworthy that on par with the secreted proteins, the proteins originated from different extracellular vesicles, and proteins probably released by OCs during the fusion process were found. *B. moojeni* venom has proteases and other components that are not yet described. While not killing the cell, the venom’s action breaks down the membrane and promotes processes causing the release of proteins that are not normally secreted or proteins related to catalytic and protein disturbances. Our data show that all three groups present a distinct pattern of the identified proteins. We selected some of these proteins for further discussion (Table 1). 

Using Panther software analysis, several categories of enrichment data were detected (Figure 3). First, we analyzed the molecular function of the identified proteins. The groups treated with *B. moojeni* crude venom and HMM proteins were related to transporting activity (9.1% and 6.3%, respectively), which cannot be found in the positive or negative controls as in the LMM-treated OCs. The venom-treated group and especially the negative control group differed from other groups by the expression of proteins with translation regulatory activity at 2.3% and 8.3%, respectively. In comparison with other groups, LMM-treated OCs did not present cell signaling protein expression (Figure 3). 

Analysis of the biological process (Figure 4) shows two different protein classes in the positive but not the negative control, such as biological adhesion and growth. The group treated with *B. moojeni* crude venom presents proteins related to the immune system (0.8%), which can also be found in the negative control (1%). The group treated with HMM fraction shows an increase in the percentage of proteins related to the immune process (2.4%) and locomotion (7.1%). In contrast, the LMM group shows fewer proteins involved in these processes. The crude venom does not present proteins involved in the multi organism, reproduction, and reproductive process.

Pathway analysis (Figure 5) indicates those proteins important to OCs differentiation in the positive control group. The Hedgehog signaling pathway (P00025), histamine H1 receptor-mediated signaling pathway (P04385), and oxytocin receptor-mediated signaling pathway (P04391) are highlighted. In the group treated with *B. moojeni* crude venom (Figure 6), the interleukin signaling pathway (P00036), inotropic glutamate receptor pathway (P00037), metabotropic glutamate receptor group III pathway (P00039), PI3 kinase pathway (P00048), and pyrimidine metabolism (P02771) are evidenced. In the group treated with HMM fraction, the PI3 kinase (P00048) and pyrimidine metabolism (P02771) pathways are of importance, while the group treated with LMM shows a single pathway involved in blood coagulation (P00011). All pathways found in the treated groups are exclusive to each group, except blood coagulation, which was observed in the group treated with LMM fraction and the negative control.

### 2.4. Secreted and Non-Secreted Proteins Identified in the Positive Control of Mature OCs

The proteins not expected to be secreted under physiological conditions were also identified on treated cells that were identified in the secretome are potentially expelled from the studied cells by signaling vesicles and exosomes. They can be found free or in vesicles, and our analysis technique allowed their identification. Thus, in the positive control of mature OCs (Table 1), we identified the cytochrome P450 studs, suggesting that it can be mediated by IL-1 b and is directly related to the activation of osteoclasts in vivo [21]. Sorting nexin 10 (Table 1), when knocked out in vivo, prevents bone loss and destruction of the joint. This protein’s deficiency does not cause a blockage in osteoclastogenesis. A functional deficiency due to F-actin rings’ malformation also shows that it is responsible for decreasing the synthesis of MMP9, CtsK, and TRAP [22]. 

The Hedgehog family of secreted proteins (Table 1) is essential for the functioning of endochondral ossification, signaling, maintenance, and skeletal tissue repair. When this protein is inhibited, it can significantly inhibit the number of bone marrow OCs. It also plays an osteoprotective role concerning bone loss caused by age [23,24].

Protocadherin belongs to the subgroup of the cadherin superfamily of homophilic cell-adhesion protein family, which does not have a known function in the OCs model. However, protocadherin 7 regulates the formation of multinucleated OCs and contributes to maintaining this tissue’s homeostasis, which is induced by RANKL. A study showed in vivo that when this molecule is depleted or decreased, it increases bone mass [25]. Semaphorin 3A is commonly secreted by neurons. However, it has an osteoprotective role, suppressing osteoclastic resorption and increasing osteoblast activity in vivo, being an important mediator of the OCs model [26]. 

### 2.5. Secreted and Non-Secreted Proteins Identified in Mature OCs after Treatment with Crude Venom and Its Components

The group treated with *B. moojeni* crude venom has many proteins with a classic effect or relationship in the regulatory components of protein regulation and transcription not yet elucidated by the literature. Interestingly, such proteins are exclusive to this group. Septins are critical regulators of osteoclastic bone resorption. In OCs treated with crude venom, we identified septin 8, which plays a fundamental role in supporting and stabilizing the cytoskeleton, with only septin 9 being studied in osteoclasts. It is known that this protein is synthesized and expressed during differentiation; when there is stabilization of the septin filaments, there is inhibition in the absorption process.

The group treated with HMM fraction presents the Kelch protein, which acts as a negative regulator of OCs differentiation [27]. Proteasomes suppress osteoclastogenesis, by regulation of NFkB, RANKL, and OPG [28]. We also identified phosphatidylinositol class IA 3-kinases (PI3Ks), which are activated by the integrin avb3 and the colony-stimulating factor 1 receptor (c-Fms). When this protein is inhibited, bone absorption deficiency occurs, which is necessary to form the functional cell borders of the osteoclast [29].

In the LMM group, we identified fibrillin, which still has a somewhat unknown role in OCs. It is only known that it influences the absorptive activity of OCs and may inhibit the expression of cathepsin K, DC-STAMP, and MMP9, causing a decrease or cancellation of absorption [30]. Additionally, in the same group, we identified the protein folliculin, which, when depleted, causes osteoporosis due to an excess of OCs, thus being an important negative regulator of OCs differentiation [31].

### 2.6. OCs Signaling Pathways Identified in the Positive Control

Among the pathways found in the positive control, when mediated by the oxytocin receptor, the receptor is present in differentiated OCs and precursors, being fundamental for the influx of calcium into the cell, thus affecting osteoclastogenesis, and increasing the number of pre-osteoclasts [32]. In this model for histamine H1 receptor-mediated signaling, which promotes osteoclastogenesis in a paracrine and autocrine way, it was discovered that histamine in vivo causes an increase in the number of osteoclasts and their precursors. In the in vitro model with histamine deficiency, there is a decreased number of OCs and decreased absorption [33]. The Hedgehog pathway plays an important role in regulating OCs and their precursors, being necessary for differentiation and absorption. When these factors are inhibited, there is a decrease in the number of OCs [23]. 

### 2.7. Osteoclastogenesis-Related Signaling Pathways Identified in OCs Treated with Crude Venom or LMM and HMM Fractions

Regarding the pathways identified in the *B. moojeni* group treated with venom, we can emphasize that, in the interleukin signaling pathway (P00036), cytokines are essential in playing a regulatory role in osteoclastogenesis, acting together as anti-osteoclastogenic and pro-osteoclastogenic modulators [5]. For the glutamate-related pathway, it was reported that glutamate-type receptors and glutamate have the function of ensuring bone impact. Inotropic glutamate receptors’ activation demonstrates regulation of the OCs phenotype in vitro and maintenance of bone mass [34,35]. The PI3K-AKT pathway, which we also found in the group treated with the venom, is related to the increase in the number of osteoclasts and cytokines, with greater precursors’ effectiveness [36]. We also identify the pyrimidine pathway; it is known that extracellular nucleotides can play a role in bone regulation, signaling, and cartilage metabolism. These can be stimulated during differentiation [37]. 

In the group treated with HMM, we detected the pathways for pyrimidine and PI3. In the group treated with LMM, only a single pathway related to coagulation was revealed, which may be influenced by thrombin, which stimulates bone resorption mediated by OCs. 

## 3. Conclusions

It is known that the venom is responsible for causing inflammation in several systems and models and triggers specific pathways and processes that can be a determinant of the OCs differentiation process [3,38,39]. Classically, this process depends on various inflammatory factors and needs a strictly balanced system, as changes in the membrane, receptors, and degradation of compounds can lead to several responses that influence differentiation [2,40,41]. *B. moojeni* venom’s effect is well known in human snakebite processes [42,43,44], which have not yet been reported in the OCs model, thus being totally exclusive and relevant data. 

We demonstrated that *B. moojeni* crude venom and LMM and HMM fractions are responsible for reducing the mature OC formation without interfering with cell viability. The venom bioactive molecules, such as metalloproteases, phospholipases, L amine oxidase acid, and serine proteases, have biological activity on cellular membranes and pathways related to inflammation [45,46,47,48] and may cause multiple effects on mature OC differentiation.

The venom and its fractions were shown to be responsible for causing some morphological and cytoskeletal changes. F-actin rings are classic OCs phenotypic characteristics [49]. Besides being involved with the cytoskeleton and the cell’s sustentation, they also present essential role functionally, including motility, conformation, and fixation for absorption [50]. We analyzed the format of rings and observed that F-actin ring formation in some treated cells was affected by crude venom and both fractions when used in the studied concentrations, suggesting a commitment of the bone resorption capacity of mature OCs. F-actin ring disruption was not observed in the positive control. We suggest that in groups treated with venom and its fractions, F-actin ring integrity disruption, in general, and our tests demonstrate that the components of *B. moojeni* venom affect processes that involve biological signals related to membrane degradation, indicative of decreased function and fission of cells, which is very important for absorption and differentiation [51,52]. 

The analysis of OCs proteins in culture medium is an important parameter not yet investigated by the scientific community. Our analysis data shows the restriction and changes in the program of protein expression in mature OCs treated with venom and its fractions and points out routes of interest that corroborate with the other analyzed parameters. Compared with untreated groups, we observed a difference in the protein expression profile, which showed exclusively proteins caused by the venom’s effect. We must consider that venom treatment has an extremely inflammatory character. The exposure to inflammatory components can naturally cause changes in the differentiation OCs. The present study indicates that treatment with crude venom, HMM, and LMM causes morphological, functional, and molecular changes in mature OCs. Further investigation of compounds derived from HMM and LMM would provide knowledge of the bioactive venom molecules responsible for these changes. 

## 4. Materials and Methods 

### 4.1. PBMCs and Osteoclast Differentiation Protocol 

PBMCs were isolated by the Ficoll–Paque density gradient centrifugation method (density 1.077 g/mL—Sigma-Aldrich®, EUA). For this, approximately 20 mL of blood were collected in tubes with sodium heparin by venipuncture in the cubital fossa of healthy male volunteers aged between 25 and 40 years old (Plataforma Brasil/CEP 1,806,596). The blood was diluted in saline (0.9%), in the proportion of 1:1. Then, this blood was placed in a conical tube containing Ficoll–Paque, in a 1:3 ratio. This material was centrifuged at 400× *g* for 20 min, without acceleration. Subsequently, the cells were washed with 2(x) in saline and resuspended in 1 mL of differentiation medium: α-MEM (Thermo Fisher Scientific, Waltham, MA, USA), pH 7.4, 10% fetal bovine serum—SFB (LGC Biotecnologia, SP, Brazil), supplemented with 25 ng/mL human M-CSF (R&D Minneapolis, MN, USA), 50 ng/mL human RANKL (R&D Minneapolis, MN, USA), 5ng/mL human TGF-β1 (R&D Minneapolis, MN, USA), and 1 μM dexamethasone (Sigma-Aldrich®, EUA). An aliquot of the cell suspension was diluted 1:1 in Trypan blue to assess the viability and count the viable cells under an optical microscope with the Neubauer chamber aid. For osteoclast differentiation tests, 6 × 10^5^ PBMCs/1.9 cm^2^ were plated and cultured in 200 µL of differentiation medium, with maintenance performed twice a week with the replacement of 50% of the medium volume (and venoms/fractions) for 15 days.

### 4.2. Venom Samples Preparation

To test molecules with OC differentiation and immunoregulation potential effects, *B. moojeni* venom, available at the biobank of the Center of Excellence for the Discovery of Molecular Targets (CENTD), was used. *B. moojeni* venom (10.0 mg/mL) was also fractionated using a 10 kDa cutting membrane, resulting in low and high molecular mass fractions.

### 4.3. Cell Viability Test—Cell Count Kit 8 (CCK8)

The cells were plated following the protocol for differentiating PBMCs into osteoclasts (item 4.1). Cell viability was assessed on the 15th day of culture, using the Cell Count Kit-8 (CCK-8, Dojindo Molecular Technologies, Kumamoto, Japan), a period corresponding to 24 h after the cells were challenged with the venom and fractions. The CCK-8 solution was added at a 1:20 ratio with incubation in a humidified environment, at a tension of 5% CO2 for 4 h, and the optical density (DO) obtained in a spectrophotometer (Quant, Bio-Tek Instruments, INC) with a wavelength of 450 nm. The results were expressed as a percentage, based on calculations correlating the samples’ optical densities with the optical density of the control and the reagent (white).

### 4.4. Phosphatase Resistant of Tartarate (TRAP) Staining 

At the end of the differentiation period, the culture medium was aspirated, and the cells were washed 3(x) with PBS. The cells were fixed by adding a solution containing 25.5% citrate solution (18 mM citric acid, 9 mM sodium citrate, 12 mM sodium chloride, and surfactant pH 3.6 ± 0.1), 66.3% acetone, and 2.9% formaldehyde for 30 s at room temperature. The TRAP solution was prepared previously (according to the manufacturer’s instructions), and heated to 37 °C. Next, cells were gently washed 3(x) with PBS, and incubated in TRAP solution for 1 h at 37 °C. The TRAP solution was aspirated, and the cells washed again 3(x) with deionized water preheated to 37 °C, and the nuclei stained with Gill Hematoxylin Solution No. 3 for 1–2 min at room temperature. The staining process was completed by washing the cells with alkaline water, showing a counter-staining, allowing an analysis of the differentiated cells under light microscopy.

### 4.5. F-Actin Ring Staining

To check the formation of F-actin rings (osteoclasts phenotypic characteristic), phalloidin was staining with fluorophore Alexa Fluor 488 (Life Technologies, Carlsbad, CA, USA), a mycotoxin from the group of phallotoxins produced by *Amanita phalloides* mushrooms. The structure has an affinity for actin filaments (COOPER, 1987). The cells were fixed with a 3.7% paraformaldehyde solution for 10 min and then washed with phosphate-saline buffer (English, Phosphate Buffered Saline—PBS) pH 7.4, and permeabilized with Triton X-100. Phalloidin staining was done at a ratio of 1:200 for 30 min. The fluorescence detection was excited/emitted at 495/518 nm, under a TSi Nikon fluorescence microscope.

### 4.6. Secretome by in Solution Digestion and Mass Spectrometry Analyses 

On the last day of OC differentiation (day 15), the supernatant was collected, followed by centrifugation with cold MeOH (volume: volume), at a maximum rotational speed. The dry samples were suspended in deionized water for protein quantification by NanoDrop. Next, ammonium bicarbonate 50 mM was added for pH regulation. Protein reduction was realized with the addition of dithiothreitol (DTT) 100 mM (dissolved in water) for 30 min at 60 °C. Next, we added iodoacetamide (IAA) (dissolved in water) 200 mM and reacted for 30 min, at room temperature, protected from light. Next, we added 100 µL of ammonium bicarbonate 50 mM, and trypsin (10 ng·μL^−1^ in 100 mM Tris-HCl, pH 8.5), in a ratio of 1:100. Digestion was carried out for 18 h, at 30 °C. Finally, we stopped the enzymatic reaction by adding 50% ACN/5% TFA. All samples were dried. Samples were dissolved in solvent A (H_2_O with 0.1% acetic acid) and injected into a C18 reverse-phase chromatography column (Supelco, 3 μm, 100 Å, 50 mm × 2.1 mm) and eluted with a gradient 5–40% of solvent B (90% acetonitrile/H_2_O with 0.1% acetic acid) for 66 min, at a constant flow of 0.2 mL/min. A PDA detector Shimadzu SPD-M20A from 200 to 500 nm monitored the eluates. The MS spectra were acquired on an IT-ToF (Shimadzu Co, Kyoto, Japan) in positive mode, interface voltage at 4.5 kV, detector voltage at 1.76 kV, interface temperature at 200 °C, and collected in the range 50–2000 m/z. The MS/MS spectrum was obtained by an argon collision and obtained in a range of 50–2000 m/z. Protein identification was performed with PEAKS studio 7.0 software using the multi-algorithmic tool InChorus (PEAKS + MASCOT) for better accuracy. The identification of proteins was based on public protein databases, using the *Homo sapiens* database. The PANTHER Classification System online was used to classify the proteins according to molecular function, biological process, and pathways.

### 4.7. Statistical Analysis

The number of osteoclasts analysis between the groups was performed by the one-way ANOVA test with Tukey’s post hoc test, using the GraphPad^®^ Prism 7.0 software (GraphPad Software Inc., La Jolla, CA, USA). Values of *p* < 0.05 were considered statistically significant.

## Figures and Tables

**Figure 1 toxins-13-00459-f001:**
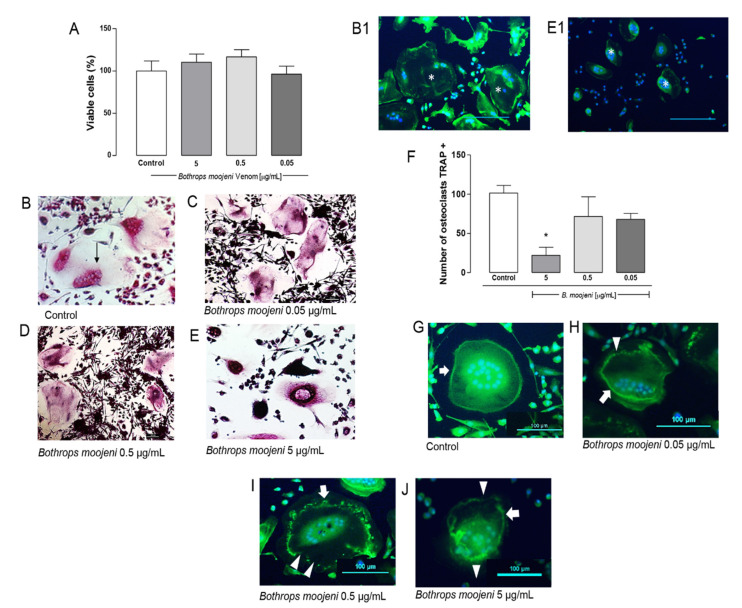
Osteoclast viability, TRAP—staining, TRAP+ OCs counting, and F-ring morphology after the treatment with *B. moojeni venom*. (**A**) CCK8 assay of mature OCs treated with crude venom viability. (**B**–**E**) OCs tartrate-resistant acid phosphatase (TRAP) staining. (**B**) TRAP+ OCs—positive control. (**C**–**E**) TRAP OCs staining after the treatment with *B. moojeni* venom at concentrations of 0.05, 0.5, and 5 µg/mL, respectively. Multinucleated TRAP+ purple cells can be observed. (**B1**) Phalloidin (green) staining, nuclei stained with DAPI (blue) of normal OCs, indicated with (white *). (**E1**) Same as in (**B1**) showing “shrunken” OCs cytoplasm, indicated with (white *), note their difference with OCs (**B1**). (**F**) Response rate curve for counting the number of TRAP + osteoclasts * *p* < 0.05. (**G**–**J**) Staining the F-actin rings with phalloidin (green), nuclei stained with DAPI (blue). OCs treated with venom at concentrations of 0.05, 0.5, and 5 µg/mL, respectively. White arrows indicate intact F-rings. White arrowheads indicate F-rings’ gradual disruption. (**H**–**J**). Scale bar: 100 µm. * *p* < 0.05 vs Control group.

**Figure 2 toxins-13-00459-f002:**
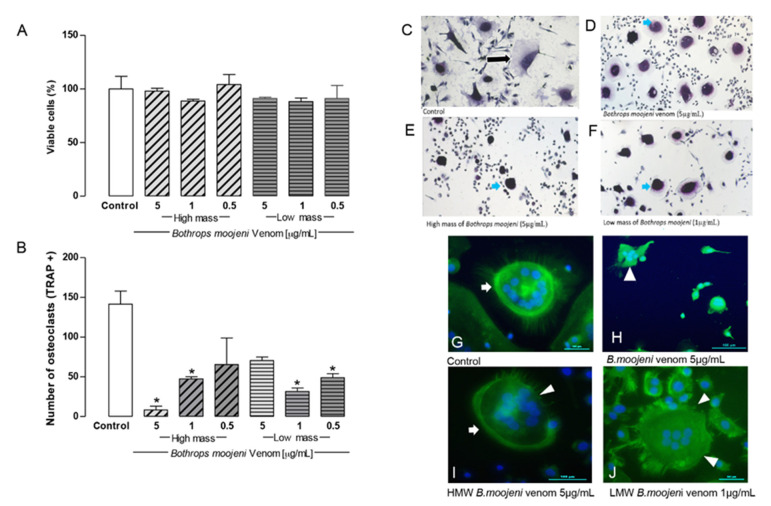
Osteoclast viability, TRAP-positive staining, TRAP+ OCs counting, and F-ring morphology after the treatment with HMM and LMM venom fractions. (**A**) Cell viability by the CCK8 method. Control group, groups treated with HMM, and LMM fractions. No toxic effect observed. (**B**) TRAP + OCs counting. Control group, groups treated with HMM, and LMM fractions, showing a significant difference in the TRAP+ OCs number. (**C**–**F**) Tartrate-resistant acid phosphatase (TRAP) staining. Culture treated with *Bothrops moojeni* venom (5 µg/mL), high mass (5 µg/mL), and low mass (1 µg/mL). (**G**–**J**) Staining of F-actin rings with phalloidin (green), nuclei stained with DAPI (blue). (**G**) An intact F-ring can be observed in the positive control. (**H**) OCs treated with crude venom (5 µg/mL), (**I**) OCs treated with HMM (5 µg/mL), and (**J**) LMM (1 µg/mL) fraction. (**H**–**J**) showed F-actin ring disruption. Blue arrows indicate differences between control (black arrow) and treated OCs. White arrows indicate intact F-rings. White arrowheads indicate F-rings’ disruption. Scale bar: 100 µm. * *p* < 0.05 vs Control group.

**Figure 3 toxins-13-00459-f003:**
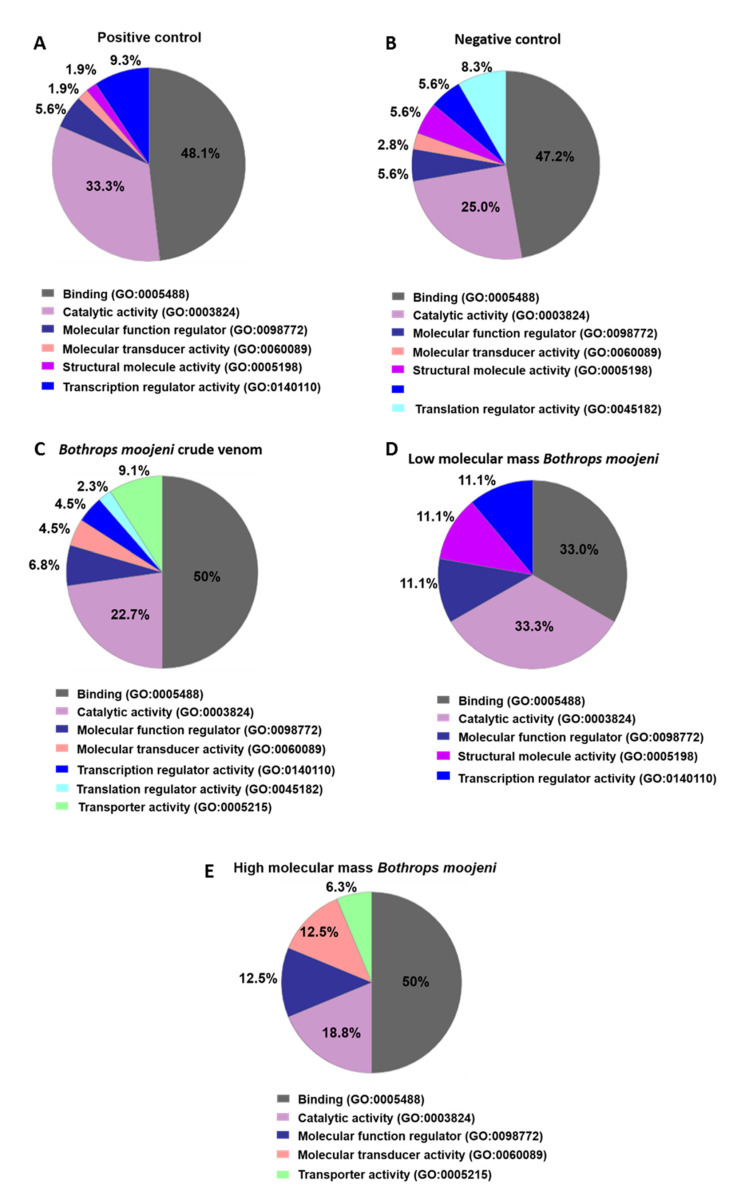
Data enrichment by molecular function using Panther software. The graphics indicate the percentage of each process. (**A**) Data enrichment graph corresponding to the positive control. (**B**) Data enrichment graph corresponding to the Negative Control. (**C**) Data enrichment graph corresponding to the *Bothrops moojeni* crude venom. (**D**) Data enrichment graph corresponding to the low molecular mass *Bothrops moojeni*. (**E**) Data enrichment graph corresponding to the high molecular mass *Bothrops moojeni*.

**Figure 4 toxins-13-00459-f004:**
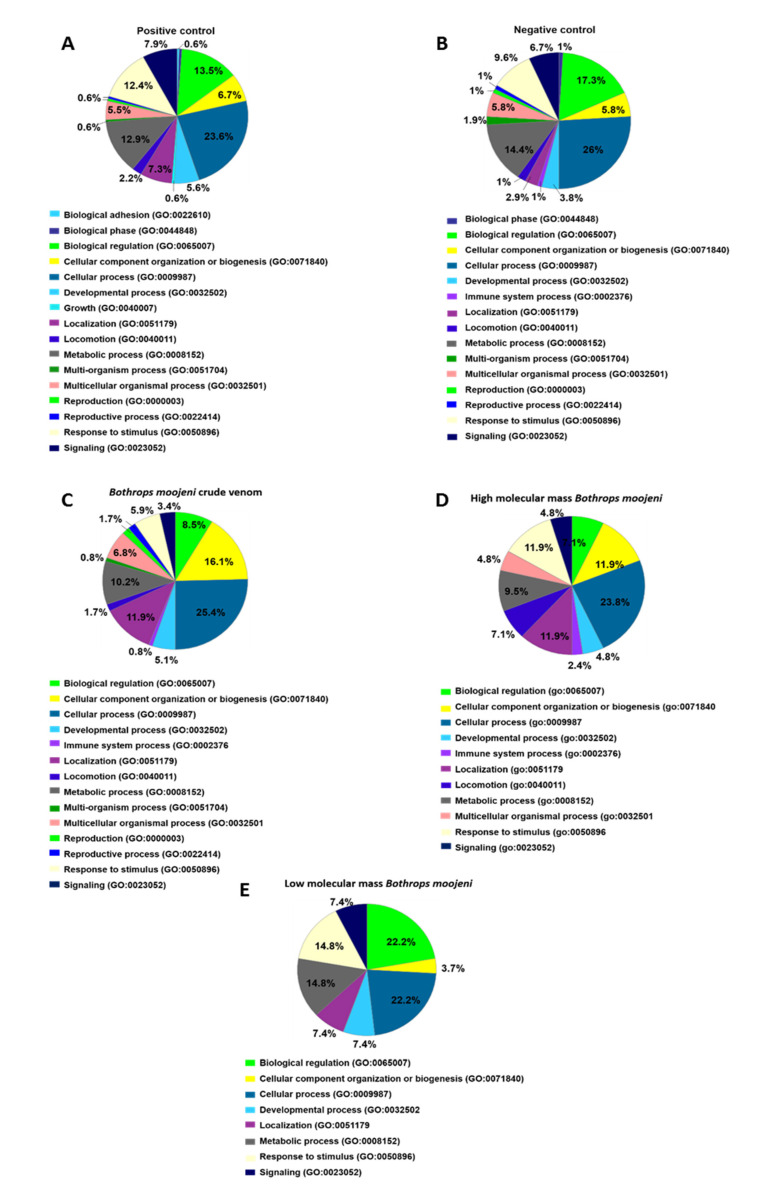
Data enrichment by biological process using Panther software. The graphics indicate the percentage of each process. (**A**) Data enrichment graph corresponding to the positive control. (**B**) Data enrichment graph corresponding to the Negative Control. (**C**) Data enrichment graph corresponding to the *Bothrops moojeni* crude venom. (**D**) Data enrichment graph corresponding to the high molecular mass *Bothrops moojeni*. (**E**) Data enrichment graph corresponding to the low molecular mass *Bothrops moojeni*.

**Figure 5 toxins-13-00459-f005:**
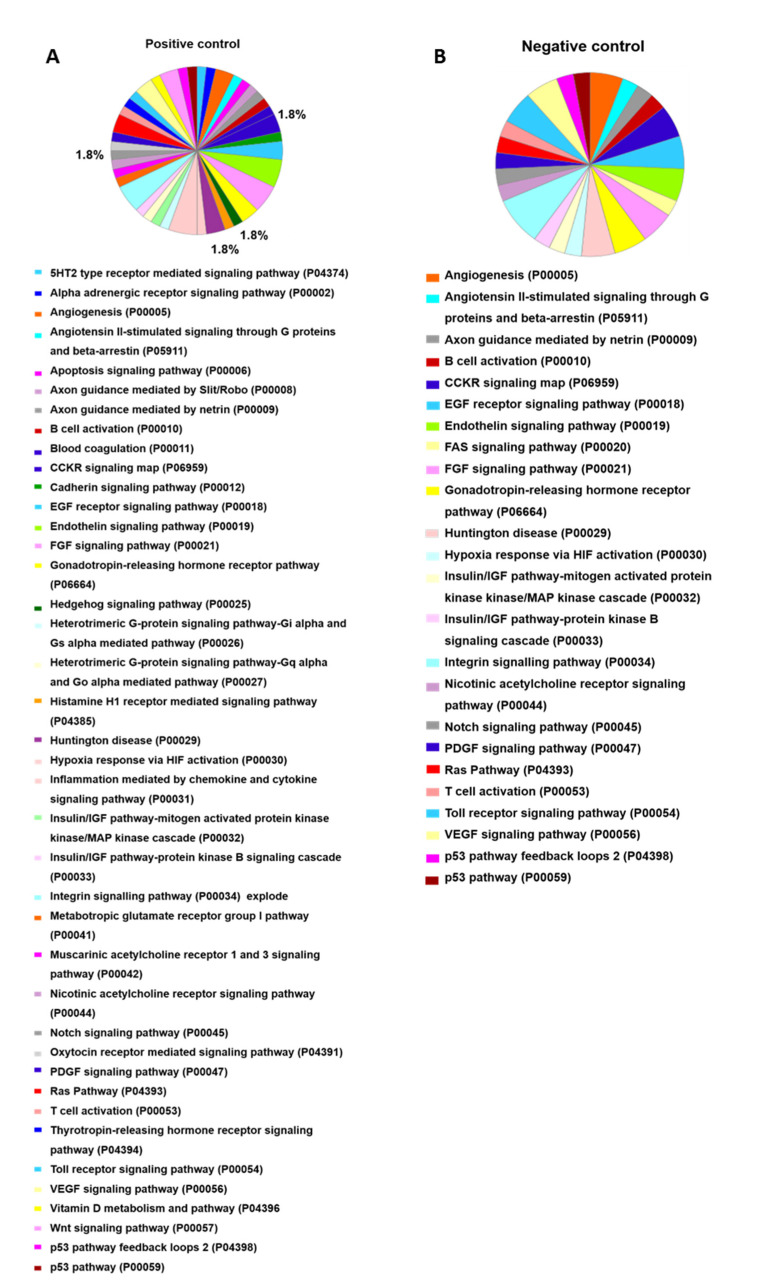
Data enrichment by pathways using Panther software. The graphics indicate the percentage of each process. (**A**) Data enrichment graph corresponding to the positive control. (**B**) Data enrichment graph corresponding to the Negative Control.

**Figure 6 toxins-13-00459-f006:**
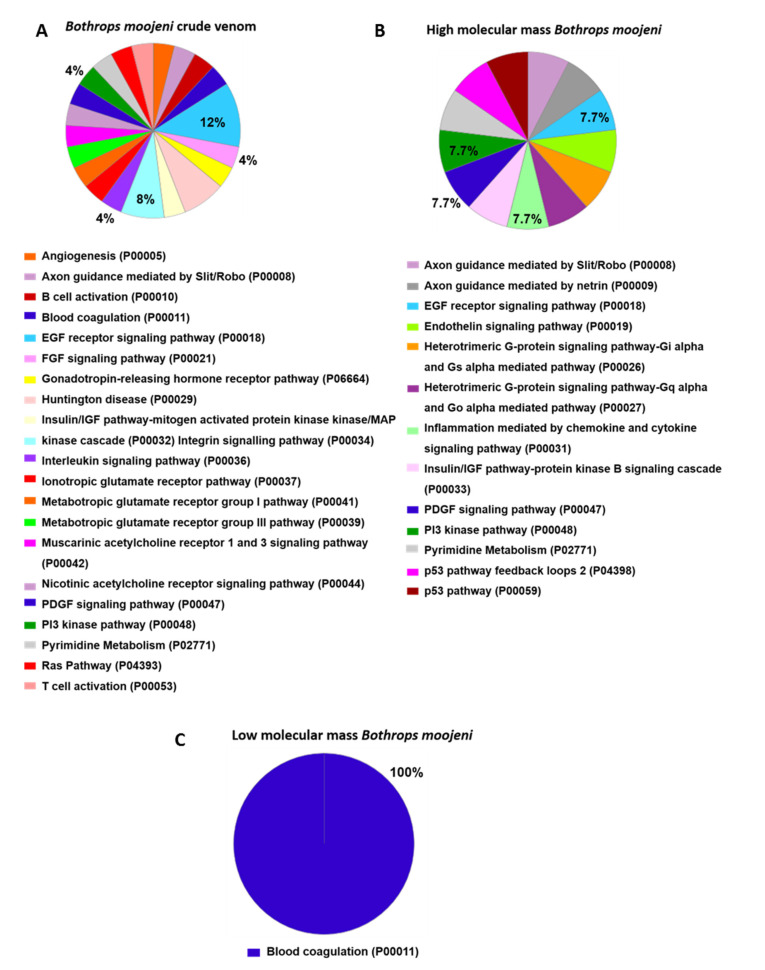
Data enrichment by biological process using Panther software. The graphics indicate the percentage of each process. (**A**) Data enrichment graph corresponding to the *Bothrops moojeni* crude venom. (**B**) Data enrichment graph corresponding to the high molecular mass *Bothrops moojeni*. (**C**) Data enrichment graph corresponding to the low molecular mass *Bothrops moojeni*.

**Table 1 toxins-13-00459-t001:** Proteins identified by mass spectrometry for the group’s positive control, negative control, *Bothrops moojeni* crude venom, low and high mass. (SP-secretory proteins; NS-non-secretory proteins, according to UniProt).

Positive Control					
Accession	Coverage (%)	Peptides	Avg. Mass	Description	Secretory
O43323|DHH_HUMAN	1	1	43,577	Desert hedgehog protein	SP
Q14623|IHH_HUMAN	1	1	45,251	Indian hedgehog protein	SP
Q15465|SHH_HUMAN	1	1	49,607	Sonic hedgehog protein	NS
Q8N118|CP4X1_HUMAN	1	1	58,875	Cytochrome P450 4X1	NS
O95025|SEM3D_HUMAN	1	1	89,651	Semaphorin-3D	SP
Q96JQ0|PCD16_HUMAN	0	1	346,182	Protocadherin-16	NS
Q96A65|EXOC4_HUMAN	1	1	110,498	Exocyst complex component 4	NS
Q9Y5X2|SNX8_HUMAN	1	1	52,569	Sorting nexin-8	NS
Negative Control					
Accession	Coverage (%)	Peptides	Avg. Mass	Description	Secretory
Q8WUM0|NU133_HUMAN	0	1	128,979	Nuclear pore complex protein Nup133	NS
O00206|TLR4_HUMAN	1	1	95,680	Toll-like receptor 4	NS
Q8N7J2|AMER2_HUMAN	1	1	69,507	APC membrane recruitment protein 2	NS
*Bothrops moojeni* Crude Venom					
Accession	Coverage (%)	Peptides	Avg. Mass	Description	Secretory
Q8N682|DRAM1_HUMAN	2	1	26,253	DNA damage-regulated autophagy modulator protein 1	NS
Q29RF7|PDS5A_HUMAN	0	1	150,830	Sister chromatid cohesion protein PDS5 homolog A	NS
Q8N3X6|LCORL_HUMAN	1	1	66,964	Ligand-dependent nuclear receptor corepressor-like protein	NS
P62805|H4_HUMAN	5	1	11,367	Histone H4	NS
P43652|AFAM_HUMAN	1	1	69,069	Afamin	SP
Q92599|SEPT8_HUMAN	1	1	55,756	Septin-8	NS
O95150|TNF15_HUMAN	3	1	28,087	Tumor necrosis factor ligand superfamily member 15	SP
Q7Z5K2|WAPL_HUMAN	1	1	132,945	Wings apart-like protein homolog	NS
High Molecular Mass of *Bothrops moojeni* Venom					
Accession	Coverage (%)	Peptides	Avg. Mass	Description	Secretory
P51686|CCR9_HUMAN	1	1	42,016	C-C chemokine receptor type 9	NS
Q2WGJ6|KLH38_HUMAN	1	1	65,541	Kelch-like protein 38	NS
Q8WYR1|PI3R5_HUMAN	1	1	97,348	Phosphoinositide 3-kinase regulatory subunit 5	NS
Q14997|PSME4_HUMAN	0	1	211,332	Proteasome activator complex subunit 4	NS
Low Molecular Mass of *Bothrops moojeni* Venom					
Accession	Coverage (%)	Peptides	Avg. Mass	Description	Secretoy
P01023|A2MG_HUMAN	1	1	163,290	Alpha-2-macroglobulin	SP
P35555|FBN1_HUMAN	0	1	312,297	Fibrillin-1	SP
Q8TF40|FNIP1_HUMAN	0	1	130,555	Folliculin-interacting protein 1	NS

## Data Availability

The data presented in this study are available within this article, its Appendix A and on request from the corresponding author.

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
