# Peer review of "Bothrops moojeni Venom and Its Components Strongly Affect Osteoclasts’ Maturation and Protein Patterns"

_toxins, 2021, doi:10.3390/toxins13070459_

Round 1
Reviewer 1 Report
Dear Authors,
Please see attached comments.
Best regards,

Author Response
The feedback was extremely important, so we appreciate all corrections and suggestions provided by the reviewers. Our work has been revised and adapted according to the suggestions, all questions have been answered, and we thank the editor and reviewers for their professional and accurate advice. We hope the revised version is now in agreement for publication.
Sincerily,
On behalf of all authors.
Reviewer 1:
The manuscript deals with the effect of B. moojeni crude venom, as well as high and low mass venom fractions on human osteoclasts (OCs). The research is important for understanding how snake venom and components affect human OCs that are important for bone maintenance, tissue regeneration et al., and are involved in inflammatory diseases. However, the manuscript has some major flaws.
- Experiment 1: Effects of crude venom and venom fragments on OCs differentiation model
It is understandable that crude venom does not affect cell viability, but venom fraction does, because venom fraction possessing the effect is diluted in crude venom when using the same concentration for both crude venom and venom fraction (Figure 1A and 1F; Figure 2A and 2F). However:
1) In Fig. 1B-1E, it looks like no difference between Fig. 1B and 1D for multinucleated and
activated OCs, as well as between Fig. 1C and 1E for “cells not competent to metabolize”.
Response: We changed Fig.1B and Fig.1F to clarify their difference with Fig.1D and Fig.1E, respectively. We hope that now our Figures became more understandable.
2) Fig. 1F, OCs treated with 0.05μg/mL is reduced more than those treated with 0.5 μg/mL, why?
Response: Fig.1F shows the small difference between number of OCs of the cells treated with 0.05μg/mL, which apparently is reduced more than those treated with 0.5 μg/mL. However, no statistical difference was observed between these two concentrations. In addition, no concentration-dependent effect was observed.
3) Fig. 1G-J: it looks like F-actin ring in Fig. 1I with 0.5 μgl/mL is more disrupted than that
with 5 μg/mL in Fig1J. No arrowheads in Fig. 1I?
Response: We add arrowheads in Fig.1I. We hope that now it is possible to see the difference between disruption and undisrupted rings.
4) Similar questions in Figure 2: No obvious differences between Fig. 2B-E.
Response: We also try to provide a more understandable picture.
5) Please clarify about “...by cutting membrane 10 kDa...” in line 10 from the bottom on page 3. Do you mean fractions with more than 10 kDa are high mass, while those with less than 10 kDa are low mass?
Response: We added a phrase to improve understanding, high mass is compounds above 10 kDa, and low mass is compounds below 10 kDa.
Reviewer suggests: 1) repeat this experiment by adding more crude venom and venom fractions, because it is hard to see the different effects of the treatments between 0.05, 0.5 and 5 μg/mL. 2) cover many cells under one view, then amplify some typical cells to show the disruptions in Figure 1B-J and Figure 2G-J.
Response: New photos were added to better elucidate the morphology, in relation to the repetition of the experiments, these doses were used because they caused a decrease in the number of osteoclasts. A higher dose would probably cause a certain toxicity, which would prevent us from seeing the cellular effect, on the other hand, the use of very low doses would also have no effect on the culture, thus being invalid for this type of study.
- Experiment 2: Secreted protein analysis
Reviewer was unable to see that how authors set up the positive and negative controls mentioned in line 9 from the bottom on page 4; so, it is hard to compare the differences of secreted proteins by five treatments (Table 1, Figure 3-6). Is OCs treated with 5 μg/mL also served as positive control? If so, then confused in lines 4-5 from the bottom on page 2.
Response: The authors set up as a negative control that which consists only of PBMCs without inducing osteoclast differentiation. Thus, the positive control is about the culture that is induced osteoclast differentiation without venom treatment, the study group is about differentiation induction and venom treatment. Being the group treated with raw venom 5, a comparative with respect to high and low mass. Sentences were added to the text to better elucidate and differentiate the groups.
Minors:
Please verify the following:
1) The first line in Introduction “... for the majorily of ...”
2) Lines 5 from bottom on page 2: “...positive control e in OCs...”, confused, please clarify.
3) Lines 3-8 from top on page 3: “... Figure 1-G... Figure 1-H... Figure 1-I.... Figure 1-J...”
4) Line 9 from bottom on page 3: “...5, 1 e 0.5 μg/mL...”
5) Line 1 from bottom on page 3: “… (Figure 2-B”.
6) Please italicize all “B. moojeni”. For example, in legend of Figure 1.
Response: We appreciate all precious Reviser 1 suggestions. We followed his corrections.
Reviewer 2 Report
Dear authors,
After a deep read of the manuscript some considerations must be taken into account.
Abstract: the last phrase is out of the context.
Introduction: The first paragraph must be moved to the final of the 4th paragraph. The last one has an information about previous results that must be moved to the discussion.
Results and Discussion: The first paragraph must be incorporated in the text. This is a loose sentence.
Although the study is ver interesting with so many results, a poor discussion was conducted. Moreover, the authors did not show in a clear manner to the readers the importance of the data obtained and the conclusion of them.
Author Response
The feedback was extremely important, so we appreciate all corrections and suggestions provided by the reviewers. Our work has been revised and adapted according to the suggestions, all questions have been answered, and we thank the editor and reviewers for their professional and accurate advice. We hope the revised version is now in agreement for publication.
Sincerily,
On behalf of all authors.
Reviewer 2:
Dear authors,
After a deep read of the manuscript, some considerations must be taken into account.
Abstract: the last phrase is out of the context.
Response: The sentence has been removed.
Introduction: The first paragraph must be moved to the final of the 4th paragraph. The last one has an information about previous results that must be moved to the discussion.
Response: The paragraphs has been rearranged.
Results and Discussion: The first paragraph must be incorporated in the text. This is a loose sentence.
Response: The paragraph has been rearranged.
Although the study is ver interesting with so many results, a poor discussion was conducted. Moreover, the authors did not show in a clear manner to the readers the importance of the data obtained and the conclusion of them.
Response: We appreciate all precious Revisor 2 suggestions. We followed his corrections.
Reviewer 3 Report
The study points out the effect of the crude Bothrops moojeni venom and low or high molecular mass fractions of this venom on osteoclasts.
The effect on cell viability and osteoclast differentiation (Figs 1 and 2) is clear. However, as shown, the secretome results are mixed. Particularly, the data of Figs 5 and 6 would be better understood if presented in tables. Even figures 3 and 4 can be improved. The numbers shown on the strips are small, and the color does not provide adequate contrast for good readability.
Also, presenting results and discussion in separate chapters would improve the comprehension of the text.
The results are interesting and fit the scope of Toxins.
In further studies, the toxin (s) responsible for this inhibitory effect on osteoclasts and the mechanism of this inhibition can be better characterized.
Author Response
The feedback was extremely important, so we appreciate all corrections and suggestions provided by the reviewers. Our work has been revised and adapted according to the suggestions, all questions have been answered, and we thank the editor and reviewers for their professional and accurate advice. We hope the revised version is now in agreement for publication.
Sincerily,
On behalf of all authors.
Reviewer 3:
The study points out the effect of the crude Bothrops moojeni venom and low or high molecular mass fractions of this venom on osteoclasts.
The effect on cell viability and osteoclast differentiation (Figs 1 and 2) is clear. However, as shown, the secretome results are mixed. Particularly, the data of Figs 5 and 6 would be better understood if presented in tables. Even figures 3 and 4 can be improved. The numbers shown on the strips are small, and the color does not provide adequate contrast for good readability.
Response: Thanks for the suggestion, we believe that the graphics are better illustrative, for what we want to draw attention to; in any case, we have improved the dimension and legend of the figure.
Also, presenting results and discussion in separate chapters would improve the comprehension of the text.
Response: Thank you for this suggestion however in our understanding, the disposition of results and discussion are better suited to our proposal.
The results are interesting and fit the scope of Toxins. In further studies, the toxin (s) responsible for this inhibitory effect on osteoclasts and the mechanism of this inhibition can be better characterized.
Response: We appreciate all precious Reviser 3 suggestions. We followed his corrections. In the future, we intend to carry out studies with molecules purified from the venom, to better understand the mechanism of action.
Round 2
Reviewer 1 Report
Thanks authors for your extensive revisions.
Reviewer 2 Report
The authors conducted the revision requested and improved the manuscript. Now it is suitable for publication.
Regards.